# S2GS: Streaming Semantic Gaussian Splatting for Online Scene Understanding and Reconstruction

**Renhe Zhang** [1]  **Yuyang Tan** [1]  **Jingyu Gong** [1]  **Zhizhong Zhang** [1]  **Lizhuang Ma** [1]  **Yuan Xie** [1]  **Xin Tan** [1]

## Abstract

Existing offline feed-forward methods for joint scene understanding and reconstruction on long image streams often perform global computation over an ever-growing set of past observations, causing runtime and GPU memory to increase rapidly with sequence length. We propose Streaming Semantic Gaussian Splatting (S2GS), a strictly causal and incremental framework that builds a 3D Gaussian semantic field from image streams without accessing future frames or reprocessing historical observations. S2GS continuously updates scene geometry, appearance, and instance-level semantics through a geometry–semantic decoupled dual-backbone design. The geometry branch performs causal modeling for incremental Gaussian updates, while the semantic branch leverages a 2D foundation vision model and a query-driven decoder to predict masks and identity embeddings. Query-level contrastive alignment and lightweight online association with an instance memory further stabilize temporal identities. Experiments show that S2GS matches or outperforms strong offline baselines, while substantially improving long-horizon scalability: it processes over 1,000 frames with much slower runtime and GPU memory growth, whereas offline global-processing baselines typically run out of memory at around 80 frames under the same setting. Project Page: https://stdcoutzrh.github.io/projects/s2gs/.

## 1. Introduction

Recently, feed-forward methods (Xu et al., 2025; Sun et al., 2025; Tian et al., 2025) built upon 3D Gaussian Splatting (3DGS) (Kerbl et al., 2023) have made substantial progress

[1]East China Normal University. Correspondence to: Xin Tan <xtan@cs.ecnu.edu.cn>.

*Proceedings of the $43^{rd}$ International Conference on Machine Learning*, Seoul, South Korea. PMLR 306, 2026. Copyright 2026 by the author(s).

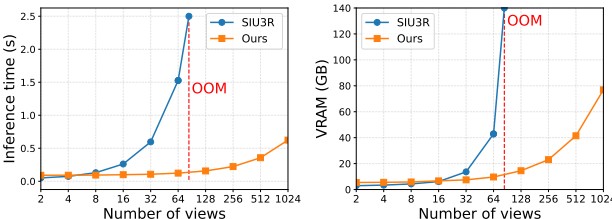

*Figure 1.* Comparison of current-frame inference time and GPU memory usage between S2GS (Ours) and the recent advanced joint reconstruction and understanding method, SIU3R(Xu et al., 2025), under varying sequence lengths in the online setting.

in jointly modeling geometry, appearance, and semantics for 3D reconstruction and scene understanding, enabling downstream applications such as robotic perception (Gong et al., 2021; Shorinwa et al., 2024; Li et al., 2025b), AR/VR (Jiang et al., 2024; Gong et al., 2026; Zhang et al., 2026), and digital twins (Craig, 2013; Jin et al., 2025). However, most existing approaches remain offline-global in the sense that, as new frames arrive, they repeatedly recompute cross-frame interactions over the growing history. While effective for short sequences, this paradigm scales poorly: both runtime and memory typically grow rapidly with the number of views, hindering long-horizon online scenarios. As shown in Figure 1, even on an H200 GPU equipped with 140 GB of VRAM, SIU3R (Xu et al., 2025) still encounters an out-of-memory (OOM) after processing approximately 80 frames, exposing a fundamental limitation of current joint modeling paradigms under long input streams. This phenomenon indicates that, for long-running online systems, there is an urgent need for an incremental modeling approach that does not require repeatedly reprocessing historical frames.

Meanwhile, recent advances (Li et al., 2025c; Huang et al., 2025; Wang et al., 2025b; Lan et al., 2025; Zhuo et al., 2025; Yuan et al., 2026) in streaming reconstruction have demonstrated better time and memory scalability. However, most existing approaches remain limited to streaming modeling of geometry and appearance, lacking semantic scene understanding and instance-level, decomposable representations, and thus falling short for downstream applications that require both reconstruction and understanding. More fundamentally, in real-world online scenarios, inputs arrive

sequentially over time and the system must update its state and produce outputs on the fly. This naturally imposes a causal constraint on online joint reconstruction and understanding: at each time step, the model can only rely on the current observation and a persistent state accumulated from the past, without access to future information or global corrections via reprocessing historical frames. Under this constraint, how to incorporate stable and temporally consistent semantic understanding while preserving the scalability of streaming inference remains an open problem. Based on the above gaps, we revisit online joint 3D reconstruction and semantic understanding for long input streams, and propose **Streaming Semantic Gaussian Splatting (S2GS)**. S2GS is designed to match the operating characteristics of real-world online systems: without repeatedly reprocessing historical frames, it incrementally maintains scene geometry, appearance, and an instance-aware semantic field, thereby unifying streaming reconstruction and semantic understanding within a single framework.

S2GS addresses two core challenges in strictly causal online joint modeling: (i) maintaining stable geometry without future-view corrections, and (ii) preserving temporally consistent instance identities under view-dependent semantic observations. To this end, we adopt reprocessing-free streaming state updates and combine geometry–semantic decoupling with identity stabilization mechanisms tailored for long-horizon inference, thereby enabling online joint reconstruction and understanding. Our contributions are summarized as follows:

1. We propose S2GS, a strictly causal and reprocessing-free framework for online joint 3D reconstruction and scene understanding, which incrementally maintains scene geometry, appearance, and an instance-level semantic field.
2. We introduce a geometry–semantic decoupled dual-backbone design tailored for streaming inference: the geometry stream performs frame-wise causal aggregation under geometric priors for stable scene maintenance, while the semantic stream independently extracts per-frame multi-scale features using a 2D foundation model, preventing geometric update noise from corrupting semantic representations.
3. We propose streaming-specific semantic and identity stabilization mechanisms: during training, query-level contrastive alignment improves cross-frame consistency, and during inference, a lightweight instance-memory association reduces ID switches; additionally, we introduce a lightweight query semantic projector with distillation to align queries to a vision–language semantic space, enabling language-conditioned query retrieval and online open-vocabulary segmentation.
4. Across multiple joint reconstruction-and-understanding benchmarks and long-horizon online settings, S2GS

achieves performance on par with or better than strong offline baselines, while significantly outperforming offline global paradigms in scalability with respect to sequence length, in terms of both inference runtime and GPU memory growth.

## 2. Related Work

**Feed-Forward 3D Reconstruction and Simultaneous Understanding.** Scene-by-scene optimization–based reconstruction paradigms (e.g., NeRF (Mildenhall et al., 2021) and 3DGS (Kerbl et al., 2023)) can deliver high-quality rendering, but optimizing a single scene often takes minutes to hours, making them ill-suited for real-time applications. Recent feed-forward reconstruction approaches have substantially accelerated the reconstruction process: both Gaussian-based methods (Charatan et al., 2024; Chen et al., 2024; Ye et al., 2024; Jiang et al., 2025) and point-map–based methods (Wang et al., 2024; Leroy et al., 2024; Wang et al., 2025a; Yang et al., 2025) demonstrate improved efficiency. However, most of these works focus primarily on recovering geometry and appearance, lacking semantic embeddings for representation, which limits their ability to further understand and parse the content of 3D environments. More recent work has begun to jointly address scene understanding and 3D reconstruction. Uni3R (Sun et al., 2025) and Uniforward (Tian et al., 2025) embed semantic features into Gaussian points, enabling zero-shot 3D semantic segmentation with arbitrary text prompts. SIU3R (Xu et al., 2025), meanwhile, uses learnable queries to endow the model with 3D understanding capability. IGGT (Li et al., 2025a) proposes an end-to-end unified Transformer that learns 3D geometry and instance-level semantics from 2D images, enabling instance-aware 3D scene understanding. Despite their ability to predict geometry/appearance and semantic information simultaneously, these approaches largely follow an offline feed-forward paradigm: given an image sequence, they repeatedly encode, match, and globally optimize over the entire sequence (including historical frames) to obtain consistent reconstruction and semantic results. This paradigm is feasible for short to medium sequences, but its computation and memory costs often grow rapidly with the number of views, directly hindering scalability to long sequences, large-scale scenes, and online real-time settings.

**Online 3D Reconstruction of Image Streams.** The streaming paradigm offers a more scalable route for 3D reconstruction. In dense 3D geometric reconstruction, several representative works have advanced along this line: Spann3R (Wang & Agapito, 2024) augments DUSt3R (Wang et al., 2024) with a memory-enhanced module to strengthen multi-view geometric fusion, while CUT3R (Wang et al., 2025b) adopts an RNN-like (Zaremba et al., 2014) incremental architec-

*Table 1.* Comparison of S2GS with prior feed-forward paradigms for 3D reconstruction and scene understanding. **CS**: causal streaming inference without using future frames; **RF**: reprocessing-free updates without repeatedly forwarding historical frames; **IS**: instance-level semantics; **SS**: streaming scalability.

| Method Category | CS | RF | IS | SS |
|---|---|---|---|---|
| Offline joint methods | ✗ | ✗ | ✓ | ✗ |
| Streaming point recon. only | ✓ | ✓ | ✗ | ✓ |
| Streaming GS recon. only | ✓ | ✓ | ✗ | ✓ |
| S2GS (Ours) | ✓ | ✓ | ✓ | ✓ |

ture that processes unstructured inputs step by step, enabling streaming inference. Going further, STream3R (Lan et al., 2025) and StreamVGGT (Zhuo et al., 2025) formulate streaming 3D reconstruction as a decoder-only causal Transformer task and leverage KV caching and causal attention to achieve scalable online reconstruction over long image sequences. For novel view synthesis, StreamGS (Li et al., 2025c) proposes an online generalizable 3D GS reconstruction method for uncalibrated image streams, enabling efficient, continuous, high-fidelity reconstruction without per-scene optimization. Although the above methods have made notable progress in streaming geometric modeling or view synthesis, they mostly focus on geometry or appearance modeling itself and do not yet fully address the joint modeling of semantic information in streaming scene reconstruction. In contrast, we propose an incremental framework for joint scene and semantic reconstruction, which updates geometric structure and semantic representations online as new images arrive, without repeatedly performing forward passes over historical observations. This enables unified modeling of scene geometry and semantics while maintaining efficient streaming inference. Online 3D reconstruction has also been extensively studied in the SLAM literature (Zhu et al., 2022; Matsuki et al., 2024; Li et al., 2024; Deng et al., 2025; Hu & Han, 2025; Sandström et al., 2025). SLAM-based methods represent another important paradigm for online reconstruction, typically relying on explicit pose tracking, keyframe management, bundle adjustment, or map optimization to incrementally update the scene. While these optimization-based procedures improve mapping accuracy, they may introduce additional computational overhead and limit reconstruction efficiency on long image streams or dense Gaussian maps. In contrast, S2GS follows a feed-forward streaming paradigm, updating geometry, appearance, and instance-level semantics in a single-pass incremental manner without repeatedly performing global computation over historical observations.

**Paradigm Comparison and Positioning.** The above feed-forward lines of work differ mainly in whether they support (i) causal streaming inference without accessing future frames, (ii) reprocessing-free updates without repeatedly

forwarding historical observations, and (iii) instance-level semantic understanding under scalable long-horizon settings. To make these capability trade-offs explicit, Table 1 summarizes representative feed-forward paradigms from this perspective. S2GS is positioned to combine the desirable properties: it performs causal, reprocessing-free online updates from RGB-only uncalibrated streams, while maintaining instance-level semantics with scalable long-horizon inference.

## 3. Method

### 3.1. Overview and Online Setting

We consider online 3D reconstruction and instance-level semantic understanding from an uncalibrated RGB video stream $\mathcal{I} = \{I_t\}_{t=1}^{T}$, where $I_t \in \mathbb{R}^{3 \times H \times W}$. S2GS operates under a strictly causal setting: at time $t$, the model processes only the current frame $I_t$ together with persistent states accumulated from previous steps, without reprocessing historical frames. The model maintains a persistent 3D Gaussian scene representation and an instance-aware semantic state, enabling scalable long-horizon streaming inference. As illustrated in Figure 2, S2GS decouples geometry and semantics into two streams: a causal geometry stream for incremental reconstruction and a semantic stream for per-frame prediction and identity maintenance.

### 3.2. Causal Transformer for 3D Gaussian Regression

**Causal Transformer with online state.** Following prior designs (Wang et al., 2025a; Lan et al., 2025), each frame is encoded into visual tokens (Oquab et al., 2023) and processed by a causal Transformer encoder. Causality is enforced by a temporal attention mask that restricts tokens at time $t$ to attend only to the history prefix $\{1, \ldots, t\}$:

$$M_{t,\tau} = \begin{cases} 0, & \tau \leq t, \\ -\infty, & \tau > t. \end{cases} \quad (1)$$

This design allows parallel processing of training clips while remaining equivalent to an autoregressive causal model. At inference, key/value tensors from past frames are cached and reused, enabling efficient long-horizon streaming without re-forwarding previous inputs.

**Incremental 3D Gaussian construction.** Under the causal constraint, the Transformer aggregates information from $\{I_\tau\}_{\tau \leq t}$ to form geometry features $H_t$. We attach three lightweight heads to predict a dense depth map $\hat{D}_t$, camera parameters $\hat{P}_t$, and per-pixel Gaussian attributes $\hat{A}_t$. To stabilize online reconstruction, we follow (Sun et al., 2025; Jiang et al., 2025) and distill geometric supervision from a strictly causal pretrained 3D foundation model (teacher) (Lan et al., 2025). Specifically, at each time step $t$, the teacher provides pseudo depth $\tilde{D}_t$ and pseudo cam-

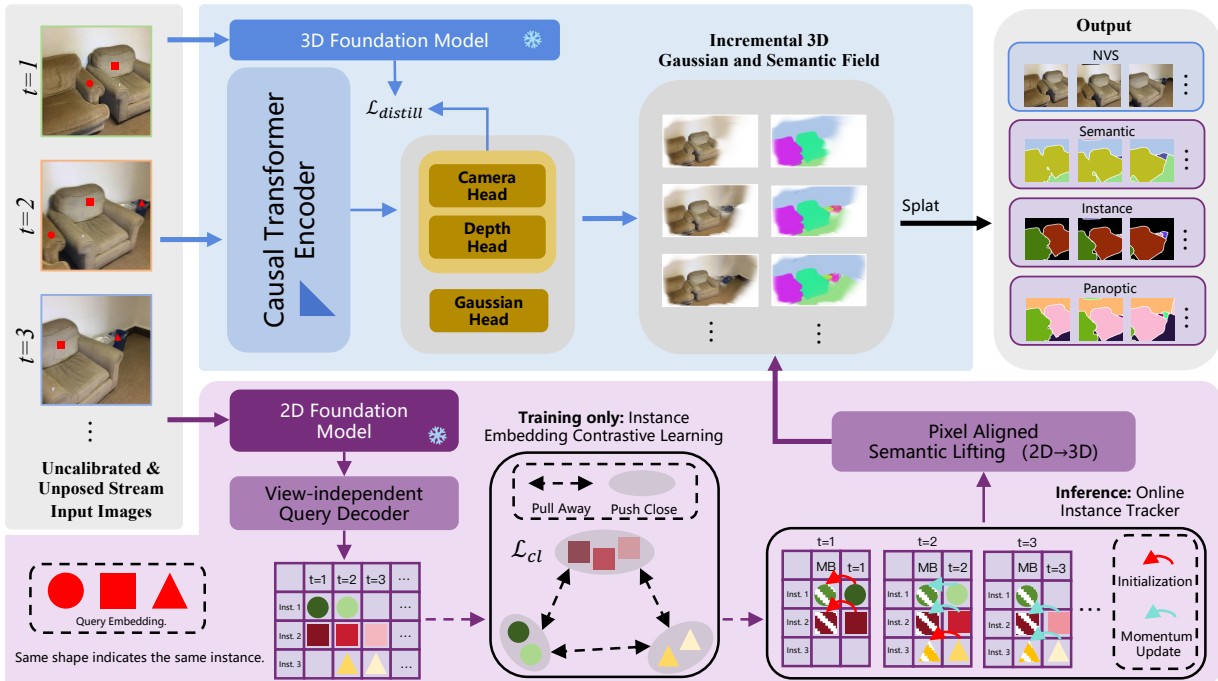

*Figure 2.* **Overview of S2GS.** S2GS processes an uncalibrated and unposed RGB image stream in a strictly causal manner. A causal Transformer encoder, guided by geometric priors from a 3D foundation model, predicts camera parameters, depth, and Gaussian attributes to incrementally construct 3D Gaussian representation. A decoupled semantic stream leverages a 2D foundation model and a query-driven decoder to produce per-view semantic and instance predictions. Query-level contrastive learning and an online instance memory bank (MB) stabilize instance identities over time. Semantic confidence is lifted to the 3D Gaussian field and decoded via splatting, enabling unified novel view synthesis, semantic segmentation, instance segmentation, and panoptic segmentation without revisiting past frames.

era/pose $\tilde{P}_t$. We supervise the student by (i) an $\ell_2$ loss on depth over valid pixels and (ii) a Huber loss on the camera parameters, encouraging the predicted geometry (depth and camera) to remain consistent with the teacher. With the distilled $\hat{D}_t$ and $\hat{P}_t$, we back-project pixels to obtain pixel-aligned 3D Gaussian centers, and combine them with the predicted attributes $\hat{A}_t$:

$$\mathcal{G}_t = \text{BackProj}\left(\hat{D}_t, \hat{P}_t\right) \oplus \hat{A}_t. \tag{2}$$

The resulting Gaussians are incrementally accumulated into a persistent global scene representation and used for differentiable rendering.

### 3.3. Online Instance Tracking and Semantic Stabilization

**Decoupled Semantic Stream with Query-based Segmentation.** To prevent interference from geometric updates, semantic representation learning is decoupled from geometric modeling. Each incoming frame $I_t$ is encoded by a frozen 2D vision foundation model (Tschannen et al., 2025) to extract robust semantic features. A lightweight adapter (Chen et al., 2022) converts these features into multiscale representations, which are consumed by a query-based

mask-classification decoder (Cheng et al., 2022). At each time step, a fixed set of learnable queries attends to the current frame to produce per-frame masks, class scores, and query embeddings:

$$\mathbf{Q}_t = \text{Dec}\left(\mathbf{Q}_0, \ \{\mathbf{F}_t^{(m)}\}\right). \tag{3}$$

The resulting query embeddings $\mathbf{Q}_t$ serve as identity descriptors for subsequent instance association.

**Online Instance Stabilization via Query-level Contrastive Learning.** During training, we align predicted queries to ground-truth instances independently for each frame using Hungarian (Kuhn, 1955) matching and apply supervised contrastive learning to the aligned query embeddings. This encourages embeddings corresponding to the same physical instance across frames to form compact clusters, while separating different instances. Concretely, let $\mathbf{z}_i$ denote a normalized query embedding with instance label $y_i$, and let $\mathcal{P}(i) = \{p \neq i \mid y_p = y_i\}$ be the positive set for anchor $i$. We optimize the supervised contrastive loss:

$$\mathcal{L}_{\text{cl}} = \sum_{i=1}^{|\mathcal{Z}|} \frac{-1}{|\mathcal{P}(i)|} \sum_{p \in \mathcal{P}(i)} \log \frac{\exp(\mathbf{z}_i^\top \mathbf{z}_p / \tau)}{\sum_{a \neq i} \exp(\mathbf{z}_i^\top \mathbf{z}_a / \tau)}, \tag{4}$$

where $\tau$ is a temperature hyper-parameter and anchors with

$|\mathcal{P}(i)| = 0$ are ignored.

At inference, per-frame predictions are associated with a lightweight instance memory bank using cosine similarity and bipartite matching:

$$A_{i,k}^t = \cos(\mathbf{z}_i^t, \bar{\mathbf{z}}_k^{t-1}), \tag{5}$$

where $\mathbf{z}_i^t$ denotes a normalized query embedding and $\bar{\mathbf{z}}_k^{t-1}$ is the prototype embedding of an existing instance. Matched prototypes are updated via exponential moving average:

$$\bar{\mathbf{z}}_k^t = \mathrm{Norm}\big((1-\alpha)\bar{\mathbf{z}}_k^{t-1} + \alpha\,\mathbf{z}_i^t\big), \tag{6}$$

yielding temporally consistent instance identities under streaming inference.

**2D-to-3D Semantic Lifting.** Per-frame semantic and instance predictions are lifted into the 3D domain by assigning them to pixel-aligned 3D Gaussians. Semantic attributes are treated analogously to appearance attributes and fused through differentiable Gaussian splatting, naturally aggregating multi-view evidence based on geometry and visibility. The resulting 3D semantic field can be rendered from arbitrary viewpoints, producing temporally and view-consistent semantic and instance predictions.

### 3.4. Language-driven Open-vocabulary Segmentation

Following SIU3R (Xu et al., 2025), we formulate language-driven segmentation as language-conditioned query retrieval to support open-vocabulary segmentation. The key difference is that SIU3R operates in an offline multi-view setting, where queries are jointly updated via omnidirectional cross-view attention, implicitly aggregating observations of the same instance across viewpoints and yielding temporally consistent instance semantics without explicit cross-frame alignment or post-processing. In contrast, we focus on streaming inference, where queries are generated per frame and updated online over time, making global multi-view interaction infeasible. Motivated by this discrepancy, we design a streaming-oriented language-conditioned query retrieval mechanism that explicitly aligns the semantic space and stabilizes temporal dynamics, enabling robust language matching under continuously updated queries.

**Query-to-semantic projection.** Query embeddings $\mathbf{Q}_t[n]$ from the semantic decoder are internal representations optimized for mask prediction and instance association, and are not guaranteed to lie in the joint vision–language semantic space of 2D foundation vision model (Tschannen et al., 2025). To bridge this gap, we introduce a lightweight Query Semantic Projector $g_\theta(\cdot)$ that maps each per-frame query embedding to the 2D foundation vision model (Tschannen et al., 2025) embedding space:

$$\mathbf{u}_{t,n} = g_\theta(\mathbf{Q}_t[n]) \in \mathbb{R}^{C_s}, \tag{7}$$

where $C_s$ denotes the SigLIP2 embedding dimension. During training, we freeze the semantic decoder and optimize only the projector parameters $\theta$. To provide a semantic teacher signal, we apply the predicted mask $\mathbf{m}_{t,n}$ to the input image and encode the masked region using the frozen SigLIP2 image encoder, yielding a normalized teacher embedding $\mathbf{v}_{t,n}$. We then align the projected query embedding $\mathbf{u}_{t,n}$ with $\mathbf{v}_{t,n}$ using a cosine regression loss, so that projected queries become directly comparable to SigLIP2 text embeddings.

**Stabilizing distillation under momentum-updated queries.** At inference, queries are updated across frames via momentum to improve instance-level stability, which induces temporal variations in query embeddings. To make the projection robust to such dynamics, we enforce *instance-level semantic invariance* during training: supervised query-level contrastive learning encourages embeddings corresponding to the same physical instance across frames to form a compact cluster, and during distillation we randomly aggregate via weighted averaging same-instance queries from different views and distill the aggregated embedding toward the same SigLIP2 (Tschannen et al., 2025) teacher representation. This explicitly accounts for embedding drift induced by momentum updates at test time.

**Language-driven query retrieval.** At test time, given a text description $r$, we obtain a normalized text embedding $\mathbf{e}_r$ using the SigLIP2 (Tschannen et al., 2025) text encoder and compute cosine similarity with the projected queries,

$$s_{t,n} = \cos(\mathbf{u}_{t,n}, \mathbf{e}_r), \quad n^\star = \arg\max_n s_{t,n}. \tag{8}$$

We return the corresponding mask $\mathbf{m}_{t,n^\star}$ as the segmentation result.

## 4. Experiments

### 4.1. Experimental Setup

**Implementation Details.** We train and validate on Scan-Net (Dai et al., 2017), using the SIU3R (Xu et al., 2025) preprocessed frames. The main difference lies in our long-sequence sampling strategy. Instead of collecting more context within a fixed viewpoint range, we construct streaming sequences by *progressively extrapolating the viewpoint*: each context frame is sampled such that its viewpoint extends beyond that of the previous frame, thereby continuously expanding the viewing range over time. Detailed sequence construction, the IoU definition, and training settings are provided in the appendix.

**Baselines and Metrics.** Since there are currently no publicly available online feed-forward 3DGS methods, we compare against representative offline approaches, including SIU3R (Xu et al., 2025), Uni3R (Sun et al., 2025), and LSM (Fan et al., 2024). We also include widely used 2D

*Table 2.* Comparison with feed-forward methods on the ScanNet (Dai et al., 2017) dataset under short-sequence inputs. "●", "†", and "⋆" denote reconstruction-only, understanding-only, and joint reconstruction-and-understanding methods, respectively.

| Method | 2 views | | | | | 8 views | | | | |
|---|---|---|---|---|---|---|---|---|---|---|
| | PSNR↑ | SSIM↑ | mIoU↑ | T-mIoU↑ | T-SR↑ | PSNR↑ | SSIM↑ | mIoU↑ | T-mIoU↑ | T-SR↑ |
| ● pixelSplat (Charatan et al., 2024) | 24.76 | 0.804 | - | - | - | - | - | - | - | - |
| ● MVSplat (Chen et al., 2024) | 23.63 | 0.784 | - | - | - | - | - | - | - | - |
| ● NoPoSplat (Ye et al., 2024) | 25.27 | 0.811 | - | - | - | - | - | - | - | - |
| † Mask2Former (Cheng et al., 2022) | - | - | 47.32 | 42.11 | 90.17 | - | - | 45.85 | 31.93 | 66.55 |
| † LSeg (Li et al., 2022) | - | - | 33.27 | - | - | - | - | 32.87 | - | - |
| ⋆ LSM (Fan et al., 2024) | 22.38 | 0.714 | 31.31 | - | - | - | - | - | - | - |
| ⋆ Uni3R (Sun et al., 2025) | 25.44 | 0.812 | 32.18 | - | - | 18.16 | 0.627 | 33.75 | - | - |
| ⋆ SIU3R (Xu et al., 2025) | **25.79** | **0.819** | 47.83 | 44.25 | 85.07 | 19.74 | 0.653 | 44.78 | 29.41 | 62.93 |
| ⋆ S2GS (Ours) | 24.90 | 0.810 | **52.35** | **44.89** | **93.73** | **20.83** | **0.685** | **49.53** | **33.34** | **82.49** |

| Method | 14 views | | | | | 32 views | | | | |
|---|---|---|---|---|---|---|---|---|---|---|
| | PSNR↑ | SSIM↑ | mIoU↑ | T-mIoU↑ | T-SR↑ | PSNR↑ | SSIM↑ | mIoU↑ | T-mIoU↑ | T-SR↑ |
| † Mask2Former (Cheng et al., 2022) | - | - | 43.32 | 25.43 | 55.96 | - | - | 41.59 | 25.15 | 40.91 |
| † LSeg (Li et al., 2022) | - | - | 29.17 | - | - | - | - | 30.46 | - | - |
| ⋆ Uni3R (Sun et al., 2025) | 16.36 | 0.583 | 31.31 | - | - | 16.74 | 0.593 | 32.17 | - | - |
| ⋆ SIU3R (Xu et al., 2025) | 17.29 | 0.591 | 37.38 | 27.37 | 50.18 | 17.82 | 0.629 | 39.98 | 29.39 | 41.24 |
| ⋆ S2GS (Ours) | **19.68** | **0.645** | **46.64** | **30.19** | **76.50** | **19.92** | **0.665** | **48.95** | **30.01** | **62.39** |

semantic segmentation baselines, LSeg (Li et al., 2022) and Mask2Former (Cheng et al., 2022). In addition, we report results for reconstruction-only Gaussian splatting baselines, including pixelSplat (Charatan et al., 2024), MVSplat (Chen et al., 2024), and NoPoSplat (Ye et al., 2024), which only support two-view inputs and thus are evaluated under the 2-view setting. For 3D reconstruction, we assess novel-view synthesis quality using PSNR and SSIM. For 3D scene understanding, we report per-frame semantic segmentation accuracy with mIoU, and cross-frame instance consistency using T-mIoU and T-SR. Detailed definitions of all metrics are provided in the appendix.

## 4.2. Results.

**Quantitative Results.** We evaluate S2GS on ScanNet (Dai et al., 2017) and compare it with offline feed-forward baselines. As shown in Table 2, under the extremely sparse 2-view setting, S2GS does not achieve the best PSNR/SSIM. This is expected, since offline baselines can exploit non-causal cross-view aggregation over the full input set to better resolve view ambiguity and occlusions when observations are highly limited. In contrast, S2GS is designed for streaming inputs and incrementally aggregates multi-view evidence as views arrive, without relying on global alignment. Consequently, it may be constrained by insufficient geometric cues at only two views. Nevertheless, as the number of input views increases (8/14/32), S2GS consistently improves and achieves strong performance in both reconstruction quality and temporal semantic/instance consistency, highlighting its effectiveness in practical streaming multi-view regimes.

*Table 3.* Comparison under Long-sequence input views.

| Views | Method | PSNR↑ | SSIM↑ | T-mIoU↑ | T-SR↑ |
|---|---|---|---|---|---|
| 64 | SIU3R | 15.37 | 0.579 | 24.41 | 26.10 |
| | S2GS | **18.71** | **0.638** | **26.71** | **45.91** |
| 256 | SIU3R | OOM | OOM | OOM | OOM |
| | S2GS | **16.37** | **0.581** | **20.46** | **21.37** |

**Visual comparison.** Figure 3 shows qualitative ScanNet results for novel view synthesis, semantic segmentation, and instance tracking. We include two synthesis examples: one with a small (early) viewpoint/temporal gap and one with a larger (late) gap. Overall, S2GS yields sharper, more geometrically consistent renderings and more stable, temporally consistent semantic and instance predictions, while baselines degrade with larger gaps.

**Longer-sequence performance.** Table 3 and Figure 4 show that for late-stage novel view synthesis, SIU3R (Xu et al., 2025) degrades at 64 views (blur and geometric instability) and fails at 256 views, while our method preserves sharper, more consistent renderings and still produces usable results. Overall, S2GS is more efficient and scalable for long-horizon streaming.

**Efficiency comparison under streaming input.** Table 4 reports the per-frame latency (processing the current view and updating the persistent state) and peak GPU memory (PGM) under online streaming input with different numbers of views. Our method maintains low per-frame runtime with only mild growth as the stream length increases, while

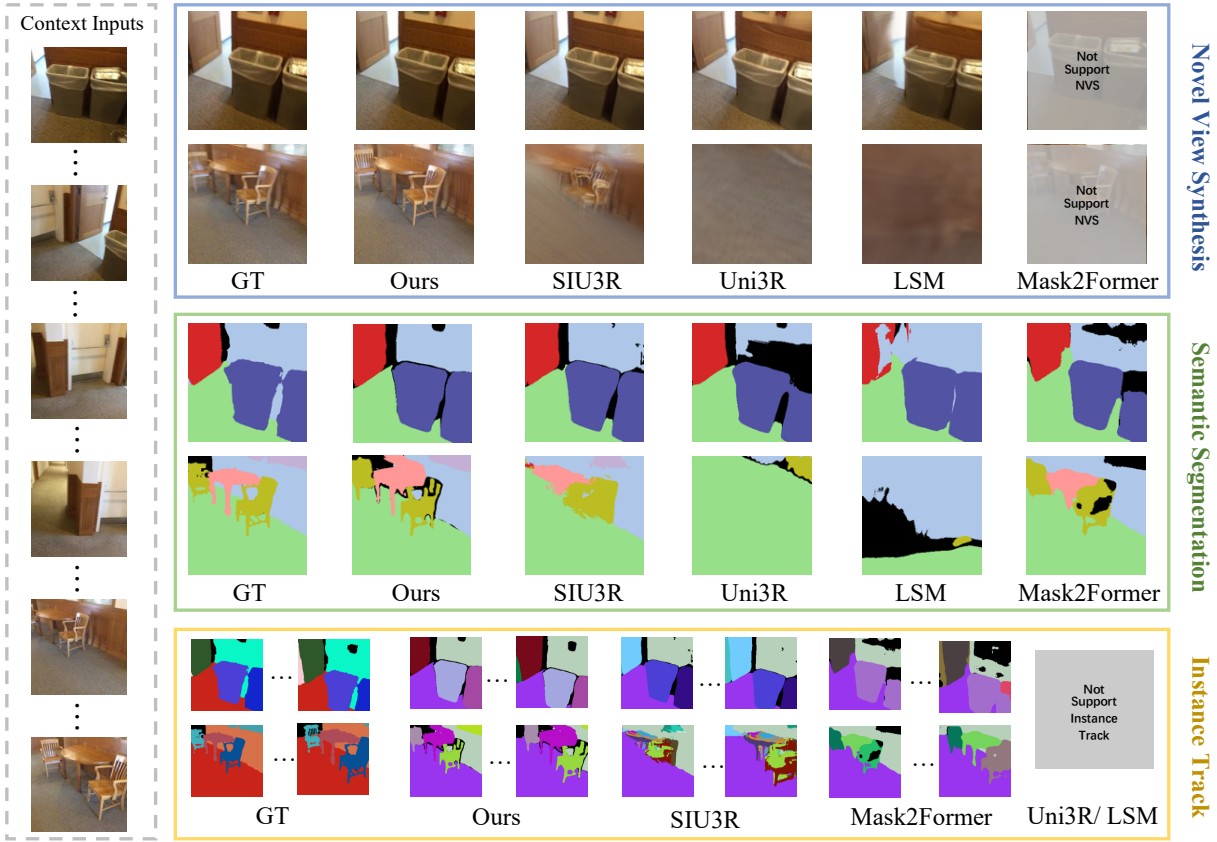

*Figure 3.* Qualitative results on ScanNet dataset.

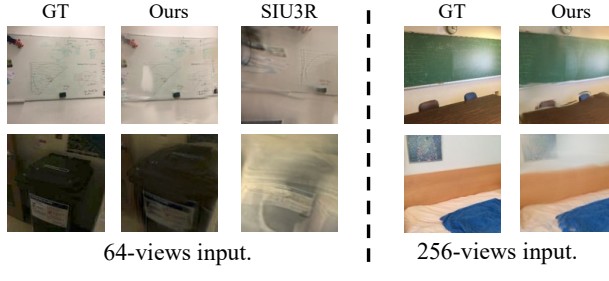

*Figure 4.* Late-stage novel view synthesis results under longer input streams.

*Figure 5.* Open-vocabulary language grounding with text queries.

SIU3R (Xu et al., 2025) exhibits rapidly growing computation and memory consumption.

**Open-vocabulary language grounding results.** We evaluate open-vocabulary grounding with text queries and visualize the results in Figure 5. As shown in Table 6, our method outperforms SIU3R (Xu et al., 2025) and LSeg (Li et al., 2022), producing more geometry-aligned masks with less bleeding. We also compare with a direct pixel-wise feature lifting (PFL) baseline that lifts dense SigLIP2 (Tschannen et al., 2025) features to 3D Gaussians. PFL obtains slightly higher mIoU by preserving dense local semantics, while

our query-based design is more suitable for instance-centric streaming, including cross-frame identity stabilization, online memory association, and panoptic reasoning. Thus, the two designs represent different trade-offs.

**Generalization evaluation.** To comprehensively assess cross-dataset generalization, we conduct zero-shot experiments on ScanNet++ (Yeshwanth et al., 2023) and Replica (Straub et al., 2019) : models are trained only on Scan-Net and directly transferred to the target datasets under the 32-view setting. The results are reported in Table 5. We

Table 4. Current-frame inference time and PGM under online streaming input.

| Views | SIU3R | | S2GS | |
|---|---|---|---|---|
| | Time(s) ↓ | PGM(GB) ↓ | Time(s) ↓ | PGM(GB) ↓ |
| 16 | 0.26 | 6.14 | 0.10 | 6.68 |
| 64 | 1.52 | 42.89 | 0.12 | 9.66 |
| 128 | OOM | OOM | 0.15 | 14.48 |
| 512 | OOM | OOM | 0.35 | 41.45 |
| 1024 | OOM | OOM | 0.62 | 76.88 |

Table 5. Zero-shot cross-dataset comparison under 32-view input.

| Method | ScanNet++ | | Replica | |
|---|---|---|---|---|
| | PSNR↑ | mIoU↑ | PSNR↑ | mIoU↑ |
| SIU3R | 12.85 | 33.51 | 13.14 | 21.42 |
| S2GS | **15.33** | **41.67** | **15.66** | **37.47** |

Table 6. mIoU of open-vocabulary language grounding from text queries.

| Method | S2GS | PFL | SIU3R | LSeg |
|---|---|---|---|---|
| mIoU | 49.37 | **51.83** | 45.76 | 34.28 |

Table 7. Comparison with Splat-SLAM (Sandström et al., 2025), a representative optimization-based online SLAM method. The subscripts $32v$ and $256v$ denote the number of input views. For Splat-SLAM, "Iters." denotes refine/init/map iterations.

| Method | Iters. | PSNR↑ | Time↓ |
|---|---|---|---|
| Splat-SLAM$_{32}$ | 1/1/1 | 9.91 | ~115s |
| Splat-SLAM$_{32}$ | 100/100/100 | **20.88** | ~345s |
| S2GS$_{32}$ | – | 19.92 | **~4s** |
| Splat-SLAM$_{256}$ | 1/1/1 | 8.82 | ~725s |
| Splat-SLAM$_{256}$ | 100/100/100 | 14.01 | ~2505s |
| S2GS$_{256}$ | – | **16.37** | ~50s |

observe that both S2GS and the baseline SIU3R (Xu et al., 2025) exhibit non-trivial zero-shot generalization capability. Nevertheless, under the same training configuration, S2GS achieves better reconstruction and semantic performance on both datasets, demonstrating stronger cross-dataset generalization and robustness.

**Comparison with Optimization-Based Online SLAM Baseline.** To better position S2GS in streaming 3D reconstruction, we additionally compare it with Splat-SLAM (Sandström et al., 2025), a representative optimization-based online SLAM method based on 3D Gaussian Splatting. As shown in Table 7, Splat-SLAM strongly depends on iterative optimization: increasing the `refine/init/map` iterations from $1/1/1$ to $100/100/100$ improves PSNR but also leads to substantially higher runtime. In contrast, S2GS is a strictly causal feed-forward method without explicit test-time optimization or revision of past states. It achieves a favorable quality–efficiency trade-off, obtaining 19.92 PSNR in ~4 s for 32 views and 16.37 PSNR in ~50 s for 256 views, compared with Splat-SLAM's 20.88 PSNR in ~345 s and 14.01 PSNR in ~2505 s under the stronger optimization setting. This comparison highlights the efficiency advantage of feed-forward streaming reconstruction, especially for longer sequences. Moreover, Splat-SLAM still requires known camera intrinsics, which may limit its applicability when such information is unavailable (e.g., web videos), whereas S2GS requires neither camera intrinsics nor camera extrinsics.

### 4.3. Ablation Studies.

**Analysis of shared backbone vs. geometry–semantic decoupling.** Our geometry–semantic decoupling is motivated by the fact that S2GS follows a 2D semantic prediction followed by 3D lifting pipeline, where the quality of 2D semantic features is critical for the final 3D semantic field. The shared-backbone variant predicts semantics from the geometry stream, while the decoupled variant uses an independent frozen SigLIP2(Tschannen et al., 2025) semantic stream to provide stronger 2D semantic representations before lifting them to 3D Gaussians. As shown in Table 8, the decoupled design substantially improves both per-frame semantic accuracy and temporal instance consistency, suggesting that a dedicated 2D semantic foundation stream is beneficial for our 2D-to-3D semantic lifting framework. Figure 6 further shows that the shared-backbone variant tends to produce blurrier semantic segmentation, which degrades instance tracking accuracy, whereas the decoupled design maintains more stable instance identities. We further equip SIU3R(Xu et al., 2025) with an independent SigLIP2-based semantic branch instead of reusing semantic features from the geometric backbone. As a result, PSNR changes marginally from 17.82 to 18.03, whereas mIoU increases substantially from 39.98 to 47.13, demonstrating that stronger semantic modeling yields significant semantic gains while incurring almost no geometric cost.

**Ablation on query-level semantic embedding contrastive learning.** Table 9 evaluates the effect of query-level contrastive learning (CL) on instance consistency under online inference. With CL, embeddings of the same instance are pulled closer while those of different instances are pushed apart, leading to a clear improvement in cross-frame instance consistency and a noticeable gain in semantic segmentation mIoU. Figure 7 further shows that CL helps the model better distinguish semantically similar instances within the

*Table 8.* Analysis of shared-backbone and geometry–semantic decoupled designs.

| Method | mIoU↑ | T-mIoU↑ | T-SR↑ |
|---|---|---|---|
| Shared | 42.17 | 21.32 | 38.75 |
| Decoupled | **48.95** | **30.01** | **62.39** |

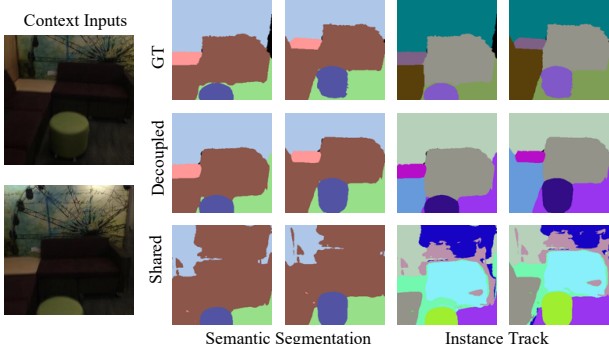

*Figure 6.* **Ablation on geometry–semantic decoupling.**

*Table 9.* Ablation study on the effectiveness of query-level semantic-embedding contrastive learning.

| Method | mIoU↑ | T-mIoU↑ | T-SR↑ |
|---|---|---|---|
| w/o CL | 47.13 | 28.64 | 50.13 |
| w CL | **48.95** | **30.01** | **62.39** |

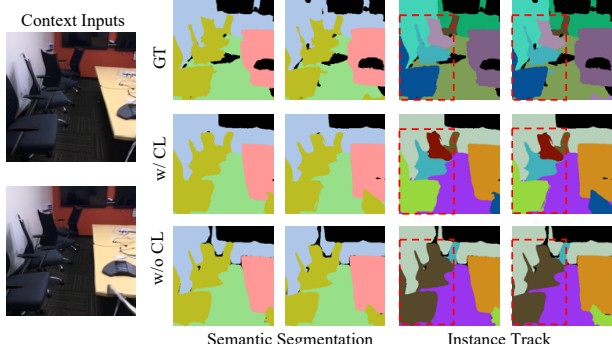

*Figure 7.* **Ablation on query-level contrastive learning.**

*Table 10.* Ablation study on geometric distillation under different sequence lengths. Distillation significantly improves reconstruction quality and becomes increasingly important for long-horizon streaming inference.

| Method | 8 views | 32 views | 64 views |
|---|---|---|---|
| w/o Distill | 15.17 | 10.21 | 10.03 |
| w Distill | **20.83** | **19.92** | **18.71** |

same category in cluttered regions (red boxes).

**Ablation study on geometric distillation.** In Table 10, we analyze the effect of geometric distillation under different sequence lengths. The distillation loss is applied only to the depth and camera heads, providing a geometric prior for stable 3D Gaussian placement during streaming inference. Without distillation, the model tends to produce superficially plausible renderings by view fitting while suffering from severe geometric drift and error accumulation. As the sequence length increases, the performance gap becomes larger, suggesting that geometric distillation not only improves short-sequence fitting but also stabilizes long-horizon streaming reconstruction.

## 5. Conclusion

We propose S2GS, a reprocessing-free framework that incrementally maintains a persistent 3D Gaussian semantic field for online scene understanding and reconstruction. S2GS decouples geometry and semantics: a causal geometry stream guided by geometric priors drives stable incremental reconstruction, while an independent semantic stream leverages 2D foundation features and a query-driven decoder to predict per-frame masks and identity embeddings. To reduce identity drift, we apply cross-frame contrastive alignment on query embeddings and perform lightweight online association with an instance memory. Experiments show that S2GS achieves performance on par with or better than strong offline baselines, while significantly outperforming offline global paradigms in scalability with respect to sequence length, in terms of both inference runtime and GPU memory growth.

## Impact Statement

This paper presents work whose goal is to advance the field of Machine Learning. There are many potential societal consequences of our work, none of which we feel must be specifically highlighted here.

## Acknowledgements

This work was supported by the National Natural Science Foundation of China (Grant Nos. 62302167, 62222602, 62502159, U23A20343, W2521174), the Shanghai Committee of Science and Technology (Grant Nos. 25511103300, 25511104302, 25511102700), the Young Elite Scientists Sponsorship Program by CAST YESS20240780, and the Open Project Program of the State Key Laboratory of CAD&CG, Zhejiang University (Grant No. A2501).

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

## Appendix

In the appendix, we provide additional content to complement the main manuscript:

- **Appendix A:** Details about Training setting

- **Appendix B:** Details about Training Objective

- **Appendix C:** Details about Metrics

- **Appendix D:** Limitations and Future Works

## A. Details about Training Setting

**Training Setup.** Unless otherwise specified, all methods are trained under the same protocol for a fair comparison. Training is conducted for 5 epochs on 8 NVIDIA H200 GPUs with a per-GPU batch size of 1 (total batch size 8). For methods that can take a variable number of input views, we train on mixed-length sequences with view counts $\{2, 8, 14\}$. At each iteration, we uniformly sample one view count and apply it to all samples in that mini-batch to avoid length-induced batch heterogeneity. We use AdamW with an initial learning rate of $1 \times 10^{-4}$ and a cosine learning-rate decay schedule. The model is optimized end-to-end with a unified objective that jointly supervises geometric/appearance reconstruction and semantic understanding.

**Data Preprocessing.** We use the ScanNet (Dai et al., 2017) frames preprocessed by SIU3R (Xu et al., 2025) . Building on these preprocessed frames, we construct our own training inputs by resampling view pairs/sequences. For two-view training, we follow SIU3R's overlap-based sampling and use the same geometric IoU computed via depth reprojection, restricting the IoU range to $[0.3, 0.8]$. For the multi-view setting, *different from SIU3R*, we build multi-frame sequences via a view-extrapolation procedure: starting from an initial frame, the $(t+1)$-th frame is sampled from later frames whose geometric IoU with the $t$-th frame lies in $[0.3, 0.8]$. Iterating this step progressively extrapolates the viewpoint, resulting in larger viewpoint changes and accumulated occlusions as the sequence grows. For the ScanNet++ (Yeshwanth et al., 2023) and Replica (Straub et al., 2019) datasets used in the zero-shot evaluation, we perform data preprocessing and sampling in a manner similar to that described above.

## B. Details about Training Objective

**Geometry and Appearance Objective.** To optimize geometry and appearance, we supervise the rendered image with both a photometric reconstruction term and a perceptual similarity term. In addition, we introduce depth distillation and camera-parameter distillation losses to further stabilize geometric structure and camera pose estimation. We also impose a rendered-depth consistency loss between the Gaussian-rendered depth and the predicted depth, and an instance-aware depth smoothness loss to encourage locally smooth geometry within the same instance region. The geometry appearance loss is defined as:

$$\mathcal{L}_{geo} = \mathcal{L}_{rgb} + \lambda_p \mathcal{L}_{lpips} + \lambda_d \mathcal{L}_{depth} + \lambda_c \mathcal{L}_{cam} + \lambda_{dc} \mathcal{L}_{depth\_consis} + \lambda_s \mathcal{L}_{depth\_smooth}, \quad (9)$$

where we set $\lambda_p = 0.05$, $\lambda_d = 0.1$, $\lambda_c = 10.0$, $\lambda_{dc} = 0.1$, and $\lambda_s = 0.05$.

**Semantic Understanding Objective.** For semantic understanding, we build upon the original segmentation-related objectives and further add a contrastive learning term to enhance representation discriminability and alignment. The semantic loss is:

$$\mathcal{L}_{sem} = \lambda_{mask} \mathcal{L}_{mask} + \lambda_{cl} \mathcal{L}_{cl}, \quad (10)$$

where $\mathcal{L}_{mask}$ is the segmentation objective derived from Mask2Former (Cheng et al., 2022). Specifically, given the predicted mask logits and the ground-truth masks, we define:

$$\mathcal{L}_{mask} = \lambda_{ce} \mathcal{L}_{ce} + \lambda_{dice} \mathcal{L}_{dice} + \lambda_{cls} \mathcal{L}_{cls}, \quad (11)$$

where $\mathcal{L}_{ce}$ is the binary cross-entropy loss on mask predictions, $\mathcal{L}_{dice}$ is the Dice loss for mask overlap, and $\mathcal{L}_{cls}$ is the classification loss for semantic categories. Following (Xu et al., 2025), we set $\lambda_{ce} = 5.0$, $\lambda_{dice} = 5.0$, and $\lambda_{cls} = 2.0$. We set the semantic mask loss weight $\lambda_{mask} = 0.05$ and contrastive loss weight to $\lambda_{cl} = 0.1$.

**Overall Training Objective.** We jointly optimize geometry–appearance and semantic understanding with the overall objective:

$$\mathcal{L} = \mathcal{L}_{geo} + \mathcal{L}_{sem}. \tag{12}$$

## C. Details about Metrics

Following prior works, we evaluate novel-view synthesis quality using PSNR and SSIM. Under the pose-free evaluation setting, we apply an alignment strategy similar to that of AnySplat (Jiang et al., 2025) to our method.

For 3D scene understanding, we report per-frame semantic segmentation mIoU and two temporal metrics (T-mIoU and T-SR) to quantify cross-frame instance consistency. For reconstruction-only methods, semantic/instance predictions are obtained on top of the 3D Gaussian Splatting (3DGS) representation and rendered to each frame, producing per-view 2D segmentation masks for evaluation.

**Per-frame mIoU.** For each frame, we compute class-wise IoU and average over $C$ semantic classes:

$$\text{IoU}_c = \frac{|P_c \cap G_c|}{|P_c \cup G_c|}, \qquad \text{mIoU} = \frac{1}{C} \sum_{c=1}^{C} \text{IoU}_c, \tag{13}$$

where $P_c$ and $G_c$ denote predicted and ground-truth pixel sets of class $c$. We then average mIoU over all frames. Since per-frame mIoU does not capture cross-frame instance consistency, we additionally report T-mIoU and T-SR (Li et al., 2025a).

**Temporal mIoU (T-mIoU).** For an object instance $o$ observed in $T$ views, with predicted masks $\{\hat{M}_t^o\}_{t=1}^{T}$ and ground-truth masks $\{M_t^o\}_{t=1}^{T}$, T-mIoU is defined as the average IoU across views:

$$\text{T-mIoU}(o) = \frac{1}{T} \sum_{t=1}^{T} \frac{\left|\hat{M}_t^o \cap M_t^o\right|}{\left|\hat{M}_t^o \cup M_t^o\right|}. \tag{14}$$

**Temporal Success Rate (T-SR).** T-SR measures whether instance $o$ is successfully tracked across all views:

$$\text{T-SR}(o) = \mathbb{I}\left[\forall t \in \{1, \ldots, T\}, \left|\hat{M}_t^o\right| > 0\right], \tag{15}$$

where $\mathbb{I}[\cdot]$ is the indicator function. Final temporal scores are averaged over all instances in the dataset.

## D. Limitations and Future Works

S2GS adopts a causal streaming setting, where inference performs incremental updates using only the current frame and a persistent state, without repeatedly forwarding historical frames. While this design provides strong online efficiency and scalability, it may also introduce error accumulation in extremely long sequences. Since the model cannot revisit historical observations for global correction, geometric reconstruction quality and semantic consistency may gradually degrade as the input stream grows. In addition, the geometric branch of S2GS partially relies on geometric priors provided by a pretrained streaming teacher model; when the teacher produces unreliable predictions, the reconstruction quality of the student model may also be affected. Future work could improve robustness by training on larger and more diverse datasets, reducing the reliance on teacher supervision, or designing lightweight causal refinement mechanisms, such as semantic-guided geometric correction, local bundle-adjustment-like state updates, or short-window consistency constraints.

Meanwhile, S2GS also inherits several common challenges in 3D Gaussian reconstruction. The current implementation does not explicitly incorporate Gaussian removal, merging, or pruning mechanisms, which may lead to redundant representations and additional computational and memory overhead in long sequences or scenes with highly redundant viewpoints. Existing strategies, such as voxelization-based Gaussian fusion, contribution-driven pruning, and budget-constrained maintenance, provide valuable references for online redundancy control. Furthermore, the current method is primarily designed for static or mildly dynamic scenes and does not explicitly model significant object motions. Future work could further explore online Gaussian maintenance mechanisms compatible with the causal streaming setting, as well as object-level dynamic modeling, to improve long-term stability and adaptability in dynamic environments.

