# OpenReview forum: "S2GS: Streaming Semantic Gaussian Splatting for Online Scene Understanding and Reconstruction"
_ICML.cc/2026/Conference — ICML 2026 regular_

### Official Review · Reviewer_4t89 · 2026-03-05

**Soundness:** 2
**Presentation:** 2
**Significance:** 2
**Originality:** 2
**Overall Recommendation:** 4
**Confidence:** 5

**Summary:**

This paper proposes S2GS, a reprocessing-free framework for streaming joint 3D reconstruction and instance-level semantic understanding from uncalibrated RGB image streams. The method decouples geometry and semantics into two streams. The geometry branch predicts depth, camera parameters, and per-pixel Gaussian attributes using a causal Transformer, while the semantic branch leverages a frozen 2D foundation model with a query-based mask decoder. Cross-frame instance consistency is maintained via supervised contrastive learning during training and a lightweight memory bank during inference. Per-frame semantic predictions are lifted to the 3D Gaussian field through pixel-aligned splatting, enabling unified novel view synthesis and semantic rendering. Experiments on ScanNet demonstrate competitive reconstruction and semantic performance compared to offline baselines, with improved scalability in long streaming sequences.

**Compliance With Llm Reviewing Policy:**

Affirmed.

**Final Justification:**

The rebuttal addresses most of my concerns, so I would like to increase my score. And I hope the content of the rebuttal should be included in the final version, if the paper can be accepted finally.

**Key Questions For Authors:**

See weakness

**Limitations:**

Yes

**Strengths And Weaknesses:**

Strength:
1. Scalable inference design. The causal Transformer with state caching avoids repeated processing of historical frames and demonstrates improved memory/runtime scaling in long sequences.
2. Practical instance stabilization mechanism. The combination of contrastive embedding learning and a lightweight memory bank provides an efficient way to maintain instance identity consistency in a streaming setting.
3. Unified 3D semantic representation. Lifting 2D semantic predictions into the Gaussian representation enables joint rendering of geometry and semantics.

Weaknesses:
1. The definition of “strictly causal” is not sufficiently clear. Although the paper claims that strictly causal inference does not reprocess historical frames or perform global corrections, Table 1 marks SLAM-based Gaussian joint methods as satisfying the SC property. In practice, SLAM systems typically rely on keyframes, bundle adjustment, and other mechanisms that revise previously estimated states, which contradicts the stated definition. This creates a conceptual inconsistency within the paper. Even if this labeling is considered an oversight, the paper does not provide a convincing justification for forbidding basic techniques such as keyframes or sliding-window optimization in its own method. These mechanisms are standard in reconstruction systems and do not necessarily cause excessive memory growth. Without explaining why such bounded-memory strategies are inappropriate, the motivation for adopting such a strong strictly causal constraint remains unconvincing.
2. The semantic reconstruction is fundamentally based on 2D predictions rather than 3D-aware modeling. Specifically, semantic masks and instance embeddings are first predicted in 2D and then lifted to 3D through geometric alignment. As a result, semantic reasoning does not explicitly leverage 3D structure or multi-view consistency during inference. This design may limit the ability to enforce cross-view semantic consistency or resolve ambiguities caused by occlusion, viewpoint changes, or incomplete observations. Errors in single-frame segmentation can be directly propagated into the 3D representation without a mechanism for global semantic refinement. Moreover, since semantic predictions are not jointly optimized in 3D space, the model may struggle in scenarios where reliable instance or category recognition requires multi-view geometric reasoning rather than per-frame 2D cues.
3. The experimental validation is limited in scope. Although it is reasonable to follow the same setting as SIU3R for fair comparison, training and evaluation primarily on ScanNet is insufficient to convincingly demonstrate robustness and generalization. The overall empirical evidence remains somewhat narrow. The limited generalization is further reflected in the results on Replica (Table 5). Replica is a synthetic dataset and generally considered less challenging than real-world data. However, the reported performance is substantially below current sota in 3D reconstruction, and in some cases even underperforms methods from several years ago (e.g., NICE-SLAM).
4. The memory consumption of the proposed method remains excessively high. Although it achieves a noticeable reduction compared to SIU3R, requiring close to 80GB of GPU memory for 1024 frames still far exceeds the hardware budget of most practical deployment scenarios. In contrast, traditional 3D reconstruction approaches (especially SLAM) are capable of processing substantially longer sequences with far lower memory usage, while often achieving stronger reconstruction quality. This comparison raises concerns about the practical scalability and efficiency of the proposed method.
5. The paper candidly acknowledges in its limitations that the current framework does not explicitly incorporate mechanisms for removing, merging, or pruning redundant Gaussians. Although S2GS demonstrates significantly improved inference runtime and GPU memory growth compared to offline global methods (as shown in Figure 1), its memory consumption still exhibits an unsaturated growth trend with respect to sequence length. This implies that the number of Gaussians accumulates approximately linearly as more frames are processed. In extremely long sequences or scenarios with highly overlapping viewpoints, this could potentially lead to degraded rendering efficiency or even memory exhaustion.
6. Why is the shared backbone inferior to the decoupled design (Table 7)?
7. SLAM GS joint methods can reconstruct scenes using only RGB sensors (Table 1).
8. In Eq. 4, the temporal scope of the positive set $\mathcal{P}(i)$ is unclear. Please clarify whether positives are constructed across time for the same instance, or only within a single frame.

---

> ### Author Rebuttal · Authors · 2026-03-31
>
> Thank you for the question and for recognizing our scalability improvement over offline feed-forward methods.We would first like to clarify that S2GS is feed-forward method, whereas SLAM methods are optimization-based;these represent two different technical paradigms.We respond to the concerns point by point below.
>
> > W1 Unclear and weakly justified strictly causal constraint
>
> As clarified above,Feed-forward methods and SLAM methods belong to different paradigms, and each paradigm includes multiple variants internally. Therefore, the categorization in Tab.1 is not sufficient to directly distinguish S2GS from SLAM methods.We will revise Tab.1 to focus on feed-forward methods only,and discuss the difference on paradigm from SLAM separately.
>
> Here “strictly causal” means old frames are not re-fed for repeated forward passes at inference, avoiding the growing cost of reprocessing history. It does not forbid lightweight state updates, Gaussian compression, or post-processing. We believe keyframes and sliding-window optimization can also help our method, but they are not our focus. Our goal is to explore whether an online method can preserve the quality of offline feed-forward methods while achieving much better scalability.
>
> > W2 2D-lifted semantics with limited 3D reasoning
>
> S2GS goes beyond 2D prediction:frame-wise semantics are lifted to pixel-aligned 3D Gaussians to form a 3D semantic field,from which novel-view semantics are rendered via geometry-aware splatting with multi-view aggregation.We validate this with two baselines.
>
> Mask2Former shares the same semantic backbone but lacks 3D semantic field modeling,and S2GS outperforms it under multiple view settings in Tab.2.
>
> We also compare with nearest-view rendering,which uses semantics from the closest source view instead of multi-view fusion.The 32-view results verify the gain from geometry-aware aggregation:
>
> |Method|mIoU↑|T-mIoU↑|T-SR↑|
> |-|-|-|-|
> |Nearest|45.78|27.17|59.68|
> |S2GS(blending)|48.95|30.01|62.39|
>
> > W3 Limited experimental scope and weak generalization
>
> Table 5 evaluates zero-shot cross-dataset transfer,rather than absolute SOTA on ScanNet++/Replica.S2GS is trained only on ScanNet and directly applied zero-shot,whereas SLAM methods still perform pose estimation and BA/graph optimization on each test scene,so the protocols are not equivalent and the absolute numbers are not directly comparable.
>
> Under the same protocol,S2GS outperforms SIU3R on both 2 datasets.We also add an auxiliary evaluation on ACID as a cross-paradigm reference,showing that feed-forward S2GS achieves a competitive performance/efficiency trade-off w/o test-time optimization.
>
> |Method|PSNR|latency|
> |-|-|-|
> |Splat-SLAM(CVPR 2025)|18.23|~336s|
> |S2GS(Ours)|17.35|~4s|
>
> > W4 Excessive memory consumption and limited practicality
>
> We agree that GPU memory is still high for very long sequences.A promising direction is KV-cache compression:we keep only the first frame as an anchor and the KV cache within a recent temporal window with window size=5.In a preliminary experiment ,this reduces VRAM from 28.03 GB to 6.71 GB,while PSNR drops only from 16.37 to 15.63,suggesting that anchor+local-window KV cache can improve practical scalability with limited accuracy loss.
>
> |Method|Anchor+Window KV Cache|Full KV Cache|
> |-|-|-|
> |PSNR|15.63|16.37|
> |VRAM|6.71|28.03|
>
> We also compare with Splat-SLAM:
>
> Splat-SLAM
>
> |Views|iters(refine/init/map)|PSNR|Time(s)|
> |-|-|-|-|
> |32|1/1/1|9.91|~115s|
> |32|100/100/100|20.88|~345s|
> |256|1/1/1|8.82|~725s|
> |256|100/100/100|14.01|~2505s|
>
> S2GS
>
> |Views|PSNR|Time|
> |-|-|-|
> |32|19.92|~4s|
> |256|16.37|~50s|
>
> SLAM methods revise past states,and their runtime/performance depend on optimization hyperparameters.By contrast,S2GS is feed-forward,requires no explicit optimization,achieves a better quality-efficiency trade-off.Moreover,Splat-SLAM still requires known camera intrinsics,which may limit its applicability when intrinsics are unavailable(e.g.,web videos),whereas S2GS requires neither intrinsics&extrinsics.
>
> > W5 No Gaussian pruning and unsaturated memory growth
>
> We agree that the current version does not explicitly prune redundant Gaussians.This limitation is valid but relatively orthogonal to our main focus.Recent work such as AnySplat explores redundancy reduction via differentiable voxelization.We will discuss deduplication as future work.
>
> > W6 Unclear advantage of the decoupled backbone
>
> The decoupled design leverages a 2D foundation model.We introduced semantic branch into SIU3R,where PSNR changes only slightly from 17.82 to 18.03 while mIoU improves from 39.98 to 47.13,showing semantic gains with little geometric cost.
>
> > W7 Inconsistent position of SLAM-GS methods
>
> As clarified in W1,we will revise Tab.1 to focus on feed-forward methods.
>
> > W8 Unclear temporal scope of the positive set in Eq.4
>
> The positive set is constructed across time:within a training clip,cross-frame query embeddings aligned to the same physical instance are treated as positive samples.

---

> > ### Author Rebuttal · Reviewer_4t89 · 2026-04-03
> >
> > Most of my concerns have been addressed, so I would like to increase my score.

---

> > > ### Author Response · Authors · 2026-04-07
> > >
> > > Dear Reviewer 4t89,
> > >
> > > Thank you for your positive feedback and for raising your score. We sincerely appreciate your time and effort in evaluating our work. We will carefully incorporate the points from the rebuttal into the final version to further improve the clarity and completeness of the paper.
> > >
> > > Best regards,
> > >
> > > Authors of Paper #692

---

### Official Review · Reviewer_jAvx · 2026-03-09

**Soundness:** 3
**Presentation:** 3
**Significance:** 4
**Originality:** 2
**Overall Recommendation:** 4
**Confidence:** 4

**Summary:**

The authors propose S2GS (Streaming Semantic Gaussian Splatting) to address the scalability limitations of joint 3D reconstruction and scene understanding methods. Existing methods typically reprocess historical frames as new data arrives, leading to rapid increases in runtime and memory usage. Experiments show that S2GS matches or exceeds offline baselines in quality while maintaining close to constant-time inference and lower memory footprints on long sequences.

**Compliance With Llm Reviewing Policy:**

Affirmed.

**Final Justification:**

I appreciate the author's rebuttal which addresses most of my concerns. Solid engineering and system design, however, the overall technical novelty is on the incremental side, I would like to keep "weak accept".

**Key Questions For Authors:**

- Does Figure 1 include the storage of the accumulating Global Gaussian Map, or just the inference buffers?
- Since geometric quality relies heavily on the Stream3R teacher, how does the system handle failure cases where the teacher drifts? Does the semantic stream have any mechanism to correct geometric errors, or is it strictly one-way?
- How does the instance memory bank handle moving objects?

**Limitations:**

- While processing memory is stable, the global map size still grows unbounded over time.

**Strengths And Weaknesses:**

Strengths:
- Scalability improvement. The most significant contribution is a strictly causal, streaming paradigm. This capability is essential for real-world applications.
- The decoupled dual-backbone architecture is well-motivated and empirically validated.
- The combination of query-level contrastive learning and an online instance memory bank is a lightweight yet effective solution.

Weaknesses:
- The geometric performance of the system is almost entirely dependent on distillation from a teacher model rather than intrinsic architectural innovation. Without this specific teacher, reconstruction quality collapses. This limits the paper's novelty claim.
- The paper claims to match or outperform offline baselines, but this is context-dependent. On standard short sequences (32 views), S2GS actually performs worse than the offline baseline SIU3R. This is an expected trade-off for a streaming system, but the abstract's claim of matching performance should refer specifically to long sequences or semantic consistency.
- One of the core claim, the ablation comparing "shared" vs. "decoupled" backbones does not control for parameter count and pre-training. The "decoupled" model benefits from a frozen, pre-trained vision-language model (SigLIP2), while the "shared" model must learn semantics from scratch using only geometric features. It's highly likely that the performance gain comes from the massive pre-training of the extra foundation model rather than the architectural decoupling itself.

---

> ### Author Rebuttal · Authors · 2026-03-31
>
> We sincerely appreciate your positive feedback on our work:scalability improvement,well motivated and effective designs.We respond to the concerns point by point below.
>
> >  Q1 Global map storage accounting
>
>  We would like to clarify that Fig.1 includes the storage of the continuously accumulated global Gaussian map, rather than only the instantaneous inference buffer. Under the strictly causal setting, S2GS maintains a persistent 3D Gaussian scene representation, and newly generated Gaussians are continuously accumulated into this global representation for subsequent rendering and updating.
>
> >  Q2 Teacher drift and error correction
>
> As discussed in the limitations, the current model still has limited ability to recover from failures caused by drift in the teacher model, and such errors may continue to propagate during incremental updates. At present, there is no geometry-correction mechanism based on the semantic stream. We believe that introducing semantic cues to detect or correct geometric errors is a highly valuable direction for future work.
>
> > Q3 Handling dynamic objects in memory bank
>
> In the current implementation, the instance memory bank uses the same association mechanism for both dynamic and static objects: each instance maintains a prototype embedding; when the object reappears in later frames, the current query is matched against the prototypes stored in the memory bank, and the matched prototype is updated with momentum. Therefore, at the identity association level, dynamic objects are not explicitly excluded or treated differently.
>
> > W1  strong teacher dependency
>
> We clarify two points.
>
> 1) S2GS initializes its geometric backbone from Stream3R.
>
> 2) Distillation is applied only to the depth and camera heads,providing a geometric prior for 3DGS point placement and preventing superficially plausible NVS results obtained by view fitting alone.
>
> Thus,geometric distillation is not a special trick for boosting performance,but a common paradigm in feed-forward/RGB-only 3D reconstruction when ground-truth geometry is unavailable. FLEG,Uni3R,and AnySplat likewise initialize from VGGT and distill depth/camera heads. Prior works also report large drops without distillation:AnySplat 18.25->7.28,FLEG 21.60->10.99.
>
> We further added distillation ablations under 8/32/64 views:
>
> |Views|w/ Distill|w/o Distill|Gain|
> |-|-|-|-|
> |8|20.83|15.17|+5.66 dB|
> |32|19.92|10.21|+9.71 dB|
> |64|18.71|10.03|+8.68 dB|
>
> These results show that distillation provides an effective geometric prior,and its benefit becomes larger on longer sequences,suggesting that it not only improves short-sequence fitting but also suppresses error accumulation and stabilizes geometry during streaming inference.
>
> At the same time,we would like to clarify that the core contribution of this work does not lie in the use of the teacher model itself,but in the proposed causal streaming joint reconstruction-and-understanding framework,together with the incremental Gaussian updating strategy,the geometry-semantic decoupling design,and the online instance stabilization mechanism developed around this setting.
>
>
> > W2 Overstated performance claim
>
> We agree that the wording in the abstract can be made more precise, but we would also like to clarify that, in the currently reported 32-view results, S2GS does not underperform SIU3R. On the contrary, under 32 views, S2GS achieves PSNR/SSIM of 19.92/0.665, higher than SIU3R’s 17.82/0.629, and also achieves better T-mIoU/T-SR, with 30.01/62.39 versus 29.39/41.24. The more noticeable trade-off in our paper actually appears under the extremely sparse 2-view setting, where offline methods can better mitigate view ambiguity through non-causal global interactions. We will tighten the wording in the revised abstract and state more explicitly that S2GS matches or surpasses strong offline baselines under practical multi-view and longer-horizon streaming settings, while offering stronger temporal consistency and scalability.
>
> > W3 Uncontrolled ablation and confounded gains
>
> We agree that the current “shared vs. decoupled” ablation is not a strictly controlled isolation experiment with matched parameter count and pretraining source, and therefore should not be over-interpreted as showing that the gain comes entirely from architectural decoupling itself. More precisely, this experiment compares two practical design choices for streaming joint modeling: one learns semantics directly from geometric-stream features, while the other introduces an independent 2D foundation semantic stream. Our results suggest that the latter is more effective under the current 2D semantic prediction → 3D lifting framework. Beyond Table 7, we also added a corresponding analysis on SIU3R: under 32 views, introducing the SigLIP2 semantic branch keeps PSNR nearly unchanged (18.03 vs. 17.82) while substantially improving mIoU (47.13 vs. 39.98).

---

> > ### Author Rebuttal · Reviewer_jAvx · 2026-04-03
> >
> > Thanks for the clarification. Given our discussion of W3, the "Ablation on shared backbone vs. geometry–semantic
> > decoupling" in the paper needs to be revised.

---

> > > ### Author Response · Authors · 2026-04-07
> > >
> > > Dear Reviewer jAvx,
> > >
> > > Thank you for your thoughtful review and positive score. We are glad that the rebuttal addressed your concerns. We will carefully incorporate your suggestions into the final revision to further improve the paper.
> > >
> > > Best regards,
> > >
> > > Authors of Paper #692

---

### Official Review · Reviewer_7iuS · 2026-03-10

**Soundness:** 3
**Presentation:** 3
**Significance:** 3
**Originality:** 2
**Overall Recommendation:** 4
**Confidence:** 3

**Summary:**

This paper proposes S2GS, a strictly causal framework for online 3D reconstruction and scene understanding from RGB video streams. The key idea is to maintain a persistent 3D Gaussian semantic field that is incrementally updated as new frames arrive, without reprocessing previous frames. The method adopts a decoupled dual-stream design: (1) a causal geometry stream that predicts depth, camera parameters, and Gaussian attributes for incremental scene reconstruction, and (2) a semantic stream that uses a frozen 2D foundation model with a query-based decoder to predict masks and instance embeddings. Experiments on ScanNet, with cross-dataset evaluation on ScanNet++ and Replica, demonstrate competitive reconstruction and semantic performance compared with prior methods.

**Compliance With Llm Reviewing Policy:**

Affirmed.

**Final Justification:**

I appreciate the author's efforts, and the rebuttal addresses most of my concerns in more baseline comparisons and implementation details. I would like to maintain my score.

**Key Questions For Authors:**

- Dependence on the teacher model. The ablation shows a gap between training with and without geometric distillation. It is unclear how much of the final reconstruction quality gain (compared to baseline methods) comes from the proposed architecture itself versus the pretrained causal teacher. Could the authors provide additional analysis to quantify the contribution of the proposed pipeline better?
- Missing baselines for 3D reconstruction. Since one contribution of the paper is enabling online Gaussian Splatting for long sequences, it would be useful to compare with closely related RGB-only implicit SLAM methods, such as NICER-SLAM and Splat-SLAM. In addition, reporting standard 3D reconstruction metrics (e.g., accuracy and completeness) would help better evaluate the geometric quality and isolate the cost of adding semantic understanding.
- Instance tracking. Can the proposed instance tracking mechanism handle objects that disappear and later reappear in the scene, or does it mainly track objects within a local temporal window until they disappear?

**Limitations:**

Yes

**Strengths And Weaknesses:**

### Strengths

- Soundness. The problem is well motivated: existing joint reconstruction-and-understanding pipelines often rely on repeated global processing over all past views, which scales poorly for long streams. The paper addresses a real limitation in online deployment through a strictly causal formulation. The decoupling of geometry and semantics is also sensible and is supported by the ablation studies.
- The paper introduces a query-based instance tracking mechanism with contrastive learning and memory-based association, which helps maintain consistent instance identities across frames in a streaming setting.
- Presentation. The paper is generally well structured and easy to follow. The comparison with prior work in Table 1 is also helpful for positioning the method.

### Weaknesses

- Originality. The novelty is somewhat incremental rather than conceptually radical. Many components build on existing techniques, including causal transformers, decoupled semantic backbones, query-based segmentation, and open-vocabulary modeling. The main contribution is their integration into a streaming Gaussian semantic field.
- The method appears to rely heavily on distillation from a pretrained model. The ablation shows that reconstruction quality drops substantially without distillation, suggesting that the student model depends strongly on teacher priors. This makes it less clear how much of the geometric improvement comes from the proposed pipeline itself.
- Geometry evaluation and missing baselines. For the 3D reconstruction task, the paper should compare with online implicit SLAM methods such as NICER-SLAM [1] and Splat-SLAM [2], which also support online reconstruction. In addition, the evaluation should include more direct 3D geometric metrics, such as accuracy and completeness. Also, some recent  SLAM + Gaussian Splatting methods use RGB-only input (e.g., Splat-SLAM [2]) suggest that the categorization in Table 1 may not be fully accurate.

[1] Zhu, Zihan, et al. "Nicer-slam: Neural implicit scene encoding for rgb slam." 2024 International Conference on 3D Vision (3DV). IEEE, 2024.
[2] Sandström, Erik, et al. "Splat-slam: Globally optimized rgb-only slam with 3d gaussians." Proceedings of the Computer Vision and Pattern Recognition Conference. 2025.

---

> ### Author Rebuttal · Authors · 2026-03-31
>
> We sincerely appreciate your positive feedback on our work:well-motivated,well structured and competitive results.We respond to the concerns point by point below.
>
> > Q1 Dependence on the teacher model
>
> We clarify two points.
>
> 1) The geometric backbone of S2GS is initialized from Stream3R weights.
>
> 2) During training, distillation is applied only to the depth and camera heads, and its essential role is to provide a geometric prior for the positions of the 3DGS points. This helps prevent the model from obtaining superficially plausible NVS results merely by fitting rendered views while neglecting the underlying true geometry.
>
> Thus,geometric distillation is not a special trick for boosting performance,but a common paradigm in feed-forward/RGB-only 3D reconstruction when ground-truth geometry is unavailable. FLEG,Uni3R,and AnySplat likewise initialize from VGGT and distill depth/camera heads. Prior works also report large drops without distillation:AnySplat 18.25->7.28,FLEG 21.60->10.99.
>
> We further added distillation ablations under 8/32/64 views:
>
> |Views|w/ Distill|w/o Distill|Gain|
> |-|-|-|-|
> |8|20.83|15.17|+5.66 dB|
> |32|19.92|10.21|+9.71 dB|
> |64|18.71|10.03|+8.68 dB|
>
> These results show that distillation provides an effective geometric prior,and its benefit becomes larger on longer sequences,suggesting that it not only improves short-sequence fitting but also suppresses error accumulation and stabilizes geometry during streaming inference.
>
> At the same time,we would like to clarify that the core contribution of this work does not lie in the use of the teacher model itself,but in the proposed causal streaming joint reconstruction-and-understanding framework,together with the incremental Gaussian updating strategy,the geometry-semantic decoupling design,and the online instance stabilization mechanism developed around this setting.
>
> > Q2 Missing baselines for 3D reconstruction
>
> We compare against Splat-SLAM (CVPR 2025) to better position S2GS in streaming reconstruction.
>
> Splat-SLAM
>
> |Views|iters(refine/init/map)|PSNR|Time|
> |-|-|-|-|
> |32|1/1/1|9.91|~115s|
> |32|100/100/100|20.88|~345s|
> |256|1/1/1|8.82|~725s|
> |256|100/100/100|14.01|~2505s|
>
> S2GS
>
> |Views|PSNR|Time|
> |-|-|-|
> |32|19.92|~4s|
> |256|16.37|~50s|
>
> SLAM methods are typically optimization-based online systems that explicitly revise past states via BA,and their runtime and performance depend strongly on optimization hyperparameters.By contrast,S2GS is feed-forward,requires no explicit optimization,and achieves a better quality-efficiency trade-off.
>
> Moreover,although Splat-SLAM is described as RGB-only,it still requires known camera intrinsics,which may limit its applicability when intrinsics are unavailable(e.g.,web videos).S2GS requires neither camera intrinsics nor extrinsics.
>
> We agree that accuracy/completeness would be useful for geometry evaluation.However,since S2GS is represented as a 3D Gaussian field rather than an explicit point cloud,fair comparison requires an additional extraction protocol that we could not include during rebuttal.As a proxy,we provide ablations with and without the semantic branch,where reconstruction quality remains nearly unchanged:
>
> |views|w|w/o|
> |-|-|-|
> |14|19.68|19.64|
> |32|19.92|19.87|
> |64|18.71|18.66|
> |256|16.37|16.31|
>
> We also added a semantic branch to SIU3R at 32 views:PSNR improves slightly from 17.82 to 18.03,while mIoU rises from 39.98 to 47.13,suggesting that semantics preserves geometry while improving semantic quality.
>
> > Q3 Instance tracking
>
> Our instance tracking is not limited to a local window.We maintain an online instance memory bank,where current-frame query embeddings are matched to stored prototypes by cosine similarity and the matched prototype is updated. Thus,objects that disappear and later reappear can in principle be linked to the same instance. The current memory bank also has no fixed capacity or stale-instance removal,so prototypes remain available for later re-identification.
>
> > W1 Many components build on existing techniques
>
> We agree that some low-level components are not entirely new;the novelty lies not in a single module,but in the problem setting and overall system design.We focus on feed-forward streaming joint reconstruction-and-understanding,which remains underexplored.More importantly,these components are not mechanically assembled,but purposefully designed around the streaming constraint and joint objective,and the ablations support their necessity.
>
> > W2 rely heavily on distillation from a pretrained model.
>
> Please refer to Q1.
>
> > W3 Geometry evaluation and missing baselines.
>
> Please refer to Q2.
>
> We also clarify that S2GS is a feed-forward streaming method,whereas SLAM methods are optimization-based online methods that explicitly refine past states(e.g.,via BA).We will revise Tab.1 to focus on feed-forward methods only,and discuss the difference from SLAM separately in the main text.

---

> > ### Author Rebuttal · Reviewer_7iuS · 2026-04-03
> >
> > I appreciate the author's efforts, and the rebuttal addresses most of my concerns in more baseline comparisons and implementation details. I would like to maintain my score.

---

> > > ### Author Response · Authors · 2026-04-07
> > >
> > > Dear Reviewer 7iuS,
> > >
> > > Thank you for your positive feedback and for maintaining your score. We are glad that the additional baseline comparisons and implementation details addressed your concerns. We will carefully incorporate these clarifications into the final version to further improve the paper.
> > >
> > > Best regards,
> > >
> > > Authors of Paper #692

---

### Official Review · Reviewer_yutB · 2026-03-13

**Soundness:** 3
**Presentation:** 3
**Significance:** 3
**Originality:** 3
**Overall Recommendation:** 4
**Confidence:** 4

**Summary:**

S2GS introduces a strictly causal, reprocessing-free framework for joint 3D reconstruction and instance-level semantic understanding from uncalibrated RGB video streams. Motivated by the scalability failure of offline feed-forward methods (which OOM at ~80 frames on H200 GPUs), S2GS maintains a persistent 3D Gaussian semantic field via incremental updates. The architecture decouples geometry (causal Transformer distilled from a frozen 3D foundation model) and semantics (frozen SigLIP2 backbone + query decoder + online instance memory). Identity stabilization is achieved via supervised contrastive learning on query embeddings during training and EMA prototype association during inference. Open-vocabulary segmentation is enabled by a lightweight query-to-SigLIP2 projector. S2GS matches or outperforms offline baselines at long horizons while using substantially less memory and runtime, and demonstrates strong zero-shot cross-dataset generalization.

**Compliance With Llm Reviewing Policy:**

Affirmed.

**Final Justification:**

Thank the authors for the additional ablation of semantic branches, time analysis and clarification of the teacher-student dependency and distillation. My concerns have been resolved. But considering that the technical novelty of integrating several existing modules, I maintain my score as "weak accept".

**Key Questions For Authors:**

**Q1**. Missing streaming baseline comparisons. Stream3R (Lan et al., 2025) is used as the teacher model for geometry distillation, yet it is not included as a reconstruction baseline. Could the authors report Stream3R's PSNR/SSIM in Tables 2 and 3? This would directly quantify the quality cost of adding the semantic stream on top of the streaming geometry backbone. Similarly, could the authors compare against StreamGS (Li et al., 2025b) on reconstruction-only metrics? This would better situate S2GS's reconstruction quality within the streaming 3DGS state of the art. Including these comparisons would significantly strengthen the paper's positioning.
**Q2**. Disentangling the teacher's contribution. Much of the geometry reconstruction quality may derive from the frozen Stream3R teacher via distillation rather than S2GS's own learned representations. Could the authors: (a) show the distillation ablation (Table 9) at multiple sequence lengths (e.g., 8, 32, 64 views) to demonstrate that distillation specifically helps with error accumulation in long-horizon inference; and (b) test with a weaker geometry teacher (e.g., CUT3R or Spann3R) to assess sensitivity to teacher quality? This analysis would help isolate S2GS's intrinsic contributions from knowledge transfer.

**Q3**. Open-vocabulary projector vs. direct SigLIP2 feature lifting. The query semantic projector maps decoder query embeddings to the SigLIP2 space for language-conditioned retrieval. An alternative baseline would be to directly lift SigLIP2 image features (per pixel or per region) to 3D Gaussians without the query decoder, and use cosine similarity for language matching. Could the authors compare against this simpler baseline to demonstrate that the query-based projector provides measurable benefit beyond direct feature lifting? Table 6 only compares against SIU3R and LSeg, not against this natural baseline.

**Q4**. Instance memory bank capacity and long-sequence identity management. The instance memory bank stores prototype embeddings for online association. What is the bank capacity? Does the system have a policy for removing stale instances (e.g., instances not seen for K frames)? In very long sequences with many instances entering and leaving, how does the bank scale? The Limitations appendix mentions error accumulation but does not specifically address instance identity management over hundreds of frames -- could the authors provide quantitative T-mIoU and T-SR results for 64-view and 128-view sequences to demonstrate temporal consistency at longer horizons?

**Limitations:**

Yes.

**Strengths And Weaknesses:**

**Soundness**: The technical design is well-grounded: the causal Transformer with masked attention correctly enforces the online constraint; the geometry distillation from Stream3R provides a principled way to inherit 3D priors without violating causality; the supervised contrastive alignment for query embeddings is a standard technique applied appropriately. The ablation studies validate each major component. Main soundness concern: the teacher-student dependency on Stream3R is not fully disentangled, making it hard to assess S2GS's intrinsic capability vs. knowledge transfer from the teacher. The distillation ablation is limited to aggregate metrics without sequence-length analysis.

**Presentation**: The paper is clearly written, well-structured, and the figures (Figs. 1, 2, 3, 4, 5) effectively communicate the key results. Table 1 providing a paradigm comparison is a useful contribution. The limitations appendix (Appendix D) is honest and concrete. Minor issues: the main-body table captions lack sufficient detail; Table 2's structure with mixed 2-view/8-view columns and methods that only support 2 views is slightly confusing.

**Significance**: The scalability problem addressed by S2GS is practically critical for long-horizon online applications in robotics, AR/VR, and autonomous navigation. Demonstrating that a streaming method can match offline quality while being orders of magnitude more scalable is a significant result. The combination of streaming reconstruction with instance-level semantics and open-vocabulary language grounding in a single causal framework fills an important gap in the literature.

**Originality**: The combination of strictly causal 3D Gaussian reconstruction with instance-level semantic understanding in a streaming setting is novel. The geometry-semantic decoupled dual-backbone design for streaming is a new architectural contribution. The streaming-specific query contrastive alignment and the query semantic projector with momentum-aware distillation are original technical contributions. The work is clearly differentiated from both offline joint methods (SIU3R, Uni3R, IGGT) and reconstruction-only streaming methods (StreamGS, Stream3R, CUT3R).

**Weakness**: Individual components in the proposed framework (causal Transformer, Mask2Former decoder, contrastive learning) are standard, without significant novelty.

---

> ### Author Rebuttal · Authors · 2026-03-31
>
> We sincerely appreciate your positive feedback on the well-grounded technical design,the originality of framework,and the well-motivated problem setting of our work.Below,we address the concerns point-by-point.
>
> > Q1 Semantic branch and streaming baselines
>
> We compared reconstruction quality w and w/o the semantic branch under 14/32/64/256 views:
>
> |views|w|w/o|
> |-|-|-|
> |14|19.68|19.64|
> |32|19.92|19.87|
> |64|18.71|18.66|
> |256|16.37|16.31|
>
> We also added a semantic branch to SIU3R at 32 views:PSNR improves slightly from 17.82 to 18.03,while mIoU rises from 39.98 to 47.13.These results suggest that the semantic branch preserves geometry while improving semantics.
>
> StreamGS is not open-sourced.We therefore compare against the online SLAM-family method Splat-SLAM(CVPR 2025)to better position S2GS in streaming reconstruction.
>
> Splat-SLAM
>
> |Views|iters(refine/init/map)|PSNR|Time|
> |-|-|-|-|
> |32|1/1/1|9.91|~115s|
> |32|100/100/100|20.88|~345s|
> |256|1/1/1|8.82|~725s|
> |256|100/100/100|14.01|~2505s|
>
> S2GS
>
> |Views|PSNR|Time|
> |-|-|-|
> |32|19.92|~4s|
> |256|16.37|~50s|
>
> SLAM methods are typically optimization-based online systems that explicitly revise past states via BA or related modules,and their runtime and performance depend strongly on optimization hyperparameters.In contrast,S2GS is feed-forward and requires no explicit optimization,achieving a better quality-efficiency trade-off.Moreover,although Splat-SLAM is described as RGB-only,it still requires known camera intrinsics,which may limit its applicability when intrinsics are unavailable(e.g.,web videos).S2GS requires neither camera intrinsics nor extrinsics.
>
> > Q2 Distillation and teacher sensitivity
>
> We clarify two points.1)the geometric backbone of S2GS is initialized from Stream3R.2)distillation is applied only to the depth and camera heads,providing a geometric prior for 3DGS point placement and preventing the model from obtaining superficially plausible NVS results by view fitting alone.This is common in feed-forward reconstruction:FLEG,Uni3R,and AnySplat similarly initialize from VGGT and distill depth/camera predictions.Prior work also reports large drops without distillation(AnySplat:18.25 -> 7.28;FLEG:21.60 -> 10.99).
>
> We added ablations under 8/32/64 views.PSNR is 20.83/19.92/18.71 with distillation versus 15.17/10.21/10.03 without.The gains are +5.66 dB,+9.71 dB,and +8.68 dB,showing that distillation not only improves short-sequence fitting but also helps suppress error accumulation in longer streaming inference.
>
> We agree this is valuable.However,in our framework,both backbone initialization and depth/camera supervision depend on the Stream3R teacher.A fair comparison would therefore require reinitializing the backbone,re-distilling,and retraining the student,rather than simply swapping the teacher at evaluation time.We will clarify in the revision that our current results rely on a strong geometric teacher prior,and that teacher-sensitivity analysis is important future work.
>
> > Q3 Open-vocabulary projector vs.direct SigLIP2 lifting
>
> Following Uni3R,we implemented a direct feature lifting baseline that lifts pixel-wise SigLIP2 features onto 3D Gaussians and performs language matching in the resulting semantic field.
>
> |Method|mIoU|
> |-|-|
> |Ours|49.37|
> |Pixel-wise feature lifting|51.83|
>
> We believe this result is reasonable.Pixel-wise lifting is better suited for dense category-level language segmentation and can preserve local semantic details more directly,whereas our method introduces an additional query projector that may create an information bottleneck.However,our query-based lifting is designed for the instance-centric streaming system:queries serve as explicit instance-level semantic carriers and are better suited for cross-frame identity stabilization,online memory association,and panoptic segmentation.We therefore view the two designs as different trade-offs for different goals,rather than a simple stronger/weaker comparison.
>
> > Q4 Instance memory bank
>
> Our current instance memory has no capacity limit and no stale-instance removal.Once written,an instance prototype is retained and updated by momentum if the instance reappears.On longer sequences,S2GS achieves T-mIoU/T-SR of 26.71/45.91 at 64 views,outperforming SIU3R’s 24.41/26.10;at 128 views,S2GS reaches 22.52/35.83.
>
> > Main soundness concern
>
> Please refer to Q2 for teacher-student dependency and distillation.
>
> > Insufficient detail in Tab.2
>
> In the revision,we will split Tab.2 into “multi-view” and “2-view-only” baselines and clarify the evaluation protocol in the caption.
>
> > Weakness:individual components are standard
>
> We agree that some low-level components are not individually new.Our contribution is not a new standalone module,but a unified solution to the underexplored problem of feed-forward streaming joint reconstruction and understanding.The framework is not a mechanical combination of existing parts;it is purpose-built for streaming joint modeling,and the ablations support this claim.

---

> > ### Author Rebuttal · Reviewer_yutB · 2026-04-06
> >
> > Thank the authors for the additional ablation of semantic branches, time analysis and clarification of the teacher-student dependency and distillation. My concerns have been resolved. But considering that the technical novelty of integrating several existing modules, I maintain my score as "weak accept".

---

> > > ### Author Response · Authors · 2026-04-07
> > >
> > > Dear Reviewer yutB,
> > >
> > > Thank you for your positive feedback. We are glad that all your concerns have been addressed and that you maintain a positive score, and we appreciate your recognition of the overall pipeline’s originality. We will carefully incorporate the additional ablations and clarifications into the final version. We believe that we all agree this work provides a unified and practical solution for feedforward streaming joint reconstruction and understanding.
> > >
> > > Best regards,
> > >
> > > Authors of Paper #692

---

### Decision · Program_Chairs · 2026-04-30

**Decision:**

Accept (regular)

**Comment:**

This paper receives 3x weak accepts. The reviewers agreed that the problem is well motivated, the proposed methods is solid in engineering and system design, and the presentation of the paper is good. One reviewer raised the final rating to weak accept as the rebuttal has clearly addressed all issues, especially regarding the experiments. Although there are still some remaining concerns on novelty of the paper, the pros from the comments of the reviews are outweighing the cons. The AC thus follows the suggestions of the reviewers to accept the paper.